Morphometric and morphologic analysis of the foramen spinosum in the Slovenian population with clinical correlations

http://orcid.org/0009-0007-2454-6592 Šink Žiga ziga.sink@mf.uni-lj.si
http://orcid.org/0000-0001-5831-2216 Umek Nejc
Cvetko Erika
Institute of Anatomy, Faculty of Medicine, University of Ljubljana , Ljubljana , Slovenia
Reno Philip
Electronic publication date: 2023 Dec 19
Publication date: 2023
Volume: 11
Electronic Location ID: e16559
Received 2023 May 19; Accepted 2023 Nov 10
Copyright: © 2023 Šink et al.
Copyright year: 2023
Copyright holder: Šink et al.
License: This is an open access article distributed under the terms of the Creative Commons Attribution License, which permits unrestricted use, distribution, reproduction and adaptation in any medium and for any purpose provided that it is properly attributed. For attribution, the original author(s), title, publication source (PeerJ) and either DOI or URL of the article must be cited.
License URL: https://creativecommons.org/licenses/by/4.0/

Keywords: Foramen spinosum, Sphenoid spine, Morphometry, Anatomical variations, Middle meningeal artery, Zygomatic root

Funding: Slovenian Research Agency P3-0043 This research was funded by the Slovenian Research Agency, Grant No. P3-0043. The funders had no role in study design, data collection and analysis, decision to publish, or preparation of the manuscript.

==============================
Background

The foramen spinosum (FS) is a pivotal passage for neurovascular structures within the skull base. We performed a detailed morphometric and morphological analysis of the FS to emphasize its clinical relevance.

Materials & Methods

The study was performed on dried skull specimens obtained from the anatomical collections of the Institute of Anatomy and Institute of Forensic Medicine of the University of Ljubljana. The morphometric and morphologic features of FS in 126 whole human skulls and 15 skull halves were analyzed, including dimensions, shape, and other anatomical variations, as well as relationships to surrounding structures. Measurements were done with a digital sliding caliper.

Results

The mean length and width of the FS were 2.45 ± 0.65 mm and 2.03 ± 0.53 mm on the right side and 2.49 ± 0.61 mm and 2.08 ± 0.48 mm on the left side. The most frequently observed shape was round (56.7%), followed by oval (28.2%), irregular (8.7%) and drop shaped (6.3%). Several anatomical variations were also noted, including FS duplication, confluences with other foramina, and FS obstruction due to marginal bony outgrowths.

Conclusion

The FS exhibits notable interindividual differences in anatomical characteristics which should be considered during neurosurgical procedures and radiological interventions in the skull base.

Introduction

The greater wing of the sphenoid bone contains three major foramina, namely the foramen ovale, the foramen rotundum and the foramen spinosum (FS), in addition to a few smaller inconsistent foramina. These foramina transmit several neurovascular structures in the skull base and may, therefore, be clinically important in cranial pathologies and surgical interventions (Standring, 2008). The FS is a circular foramen in the greater wing of the sphenoid bone situated posterolateral to the foramen ovale and anteromedial to the spine of the sphenoid bone. It connects the middle cranial fossa to the infratemporal fossa and transmits the middle meningeal artery (MMA), the middle meningeal vein connecting the cavernous sinus with the pterygoid venous plexus, and the meningeal branch of the mandibular nerve (nervus spinosus) (Standring, 2008). Its location renders it useful in various diagnostic and therapeutic procedures, especially endovascular embolization of dural arteriovenous fistulas and meningiomas through the MMA (Mewada et al., 2016). Besides being implicated in dural arteriovenous fistulas and meningiomas, the MMA has been implicated in other conditions such as true aneurysms, pseudoaneurysms, traumatic arteriovenous fistulas, moyamoya disease, recurrent chronic subdural hematomas, and migraine (Yu et al., 2016).

Accordingly, the structural characteristics of the FS bear remarkable clinical significance. Its absence is linked to the anomalous formation of the MMA (Low, 1946; Manjunath, 2001). It is a well-defined surgical landmark with proximity to critical neurovascular structures; however, its detailed anatomy has yet to be thoroughly characterized. The limited knowledge of the spatial anatomy of the FS may represent a challenge during surgery, particularly when tumors distort or invade neural structures such as the trigeminal or greater superficial petrosal nerves, making identification of the FS and nearby neurovascular structures difficult (Krayenbühl, Isolan & Al-Mefty, 2008). It is essential for surgeons to have a thorough understanding of the anatomical variations in the cranial foramina and their potential impact on associated neurovascular structures. These variations are thought to be prevalent with potential differences in different populations, but specific population data with comprehensive anatomical characterization are often lacking. Insufficient knowledge may lead to misinterpretation of computed tomography (CT) scans during the clinical evaluation of the middle cranial fossa (Sugano et al., 2022).

Fractures in the middle cranial fossa frequently occur in accidental injuries, with head trauma from road traffic accidents or direct impacts on the temporal bone potentially causing MMA disruption at the FS, leading to epidural hematoma formation (Roski et al., 1982; Chmielewski, Skrzat & Walocha, 2013). Therefore, a more comprehensive understanding of the FS and its relationship to surrounding structures could aid in the middle cranial fossa surgical approaches and trauma surgery, where exploration of the foramen may be necessary to achieve proper haemostasis (Krayenbühl, Isolan & Al-Mefty, 2008).

The primary objective of the present study was to evaluate and compare the morphometric and morphological characteristics of the FS in adult human skulls of the Slovenian territory with previously documented findings, and emphasize potential clinical significance.

Material and Methods

In this study, we performed an analysis of 266 foramen spinosum (FS) in 126 fully desiccated adult human skulls and 15 skull halves (three right-sided and 12 left-sided) of undetermined sex and age. The skulls were obtained from bodies donated by inhabitants of the Republic of Slovenia between 1965 and 2020 to the anatomical collection of the Institute of Anatomy and the Institute of Forensic Medicine at the Faculty of Medicine, University of Ljubljana, Slovenia. The skulls within the collection are systematically catalogued and assigned numerical identifiers ranging from 1 to 74, 100 to 136, and ISM1 to ISM30. Skulls exhibiting discernible damage to the relevant structures, as verified through magnified inspection, were excluded from the study. The National Medical Ethics Committee of the Republic of Slovenia granted ethical approval for this study (Permit No. 0120-459/2018/3).

The methodological approach for the evaluation of the FS was adapted from our previous study on the foramen ovale (Šink et al., 2023). Specifically, the FS located on the greater wings of the sphenoid bone were visualized and measured from both extracranial and intracranial perspectives of the cranial base. A slender wire was utilized to verify the patency of foramina and to exclude the presence of deceptive passages. The dimensions of the FS were ascertained along the anteroposterior (length, longest axis) and transverse (width, shortest axis) diameters using a digital vernier caliper with a precision of 0.01 mm. The spatial relationships between the FS and other anatomical landmarks (foramen ovale, external aperture of the carotid canal, anterior border of the zygomatic root of the temporal bone, inferior part of the zygomaticotemporal suture) were quantified employing the same methodology (Fig. 1). The shape of the FS and its potential anatomical variations (marginal bony outgrowths, divisions, duplications, confluences) were carefully recorded and photographed with a digital camera (Nikon D5600, Tokyo, Japan). Each morphometric and morphological parameter was measured or evaluated twice by a minimum of two distinct investigators, and the arithmetic mean was employed for analysis to mitigate measurement error and bias. A third investigator reviewed any inconsistent descriptions or measurements, and a consensus was reached among all authors through discussion. Prior studies were consulted to standardize assessment methodologies and anatomical nomenclature (Rai, Gupta & Rohatgi, 2012; Tewari et al., 2017; Worku & Clarke, 2021; Sugano et al., 2022).

Figure 1 Measurements collected for the study.

1. Distance between the foramen spinosum and the foramen ovale. 2. Distance between the foramen spinosum and the external aperture of the carotid canal. 3. Distance between the foramen spinosum and the anterior border of the zygomatic root of the temporal bone. 4. Distance between the foramen spinosum and the inferior part of the zygomaticotemporal suture.

For the statistical analysis, we employed the GraphPad Prism 9 software suite (GraphPad Software, San Diego, CA, USA). Data are presented as arithmetic means (standard deviation) or frequencies (proportion). A paired sample t-test and Wilcoxon rank-sum test were utilized to analyze differences between the right and left sides. A nonparametric χ2 test was employed to detect differences between proportions. The Shapiro-Wilk test was performed to evaluate the normality of the distributions. Differences were deemed statistically significant at p < 0.05.

Results

The FS was absent unilaterally in 10 out of 266 analyzed skull sides, on the right side in six out of 137 analyzed skull sides (4.38%), on the left side in four out of 129 analyzed skull sides (3.10%). In five cases, the groove of the MMA was connected with the groove of the ophthalmic artery, and in five cases, the exact origin of the MMA could not be identified. One duplication of the FS was observed on the right side (0.73%). The mean anteroposterior diameter or length of the FS was 2.45 mm on the right side and 2.49 mm on the left side. The mean transverse diameter or width of FS was 2.03 mm on the right side and 2.08 mm on the left side. The morphometric features of the FS are summarized in Table 1. No statistically significant differences were found in any measured parameter between the left and right sides.

Table 1 Morphometric data on the foramen spinosum (FS).

Parameter	Mean ± SD (mm)	Range (mm)	
Right side	Left side	Right side	Left side	
n = 130	n = 122	n = 130	n = 122	
Length of FS	2.45 ± 0.65	2.49 ± 0.61	0.45–4.17	0.82–3.97	
Width of FS	2.03 ± 0.53	2.08 ± 0.48	0.42–3.56	0.77–3.42	
Distance to FO	3.00 ± 1.29	2.99 ± 1.09	0.25–7.72	0.72–5.98	
Distance to CC	8.02 ± 1.35	8.22 ± 1.47	5.13–13.60	5.04–14.97	
Distance to AZR	21.07 ± 1.36	20.84 ± 1.29	17.24–26.22	17.05–23.77	
Distance to ZTS	35.32 ± 2.66	35.35 ± 2.68	29.61–45.11	29.58–44.83	
Note:

FS, foramen spinosum; FO, foramen ovale; CC, external aperture of the carotid canal; AZR, anterior border of the zygomatic root of the temporal bone; ZTS, inferior part of the zygomaticotemporal suture; SD, standard deviation.

The most frequently observed shape was round (56.7%), followed by oval (28.2%), irregular (8.7%) and drop shaped (6.3%). Two irregular pinhole-like FS were discovered (0.8%), both on the left side. The different FS shapes noted in the present study are shown in Fig. 2, while the classification and distribution of FS shapes are summarized in Table 2. There were no statistically significant differences between the left and right sides.

Figure 2 Shapes of the foramen spinosum (FS).

All images were captured from the external aspect of the cranial base. In each image, the white arrow points to the foramen spinosum. The red line denotes the length of the foramen spinosum and the white line denotes the width of the foramen spinosum (B). The most frequently observed shape was round (A), followed by oval (B), irregular (E, F) and drop shaped (C). The pinhole shape (D) denotes a round or an irregular foramen spinosum with a very small diameter. FS−foramen spinosum. FO−foramen ovale. FL−foramen lacerum. SS−spine of the sphenoid bone.

Table 2 Distribution of foramen spinosum (FS) shapes.

Shape of FS	Round (%)	Oval (%)	Drop (%)	Irregular (%)	
Right side (n = 130)	63.1	21.5	7.7	7.7	
Left side (n = 122 )	50.0	35.2	4.9	9.8	
Overall proportion (n = 252)	56.7	28.2	6.3	8.7	

The confluence of the FS with the foramen ovale was observed in two skull sides (1.55%) on the left side (Fig. 3). Likewise, the confluence of the FS and the foramen lacerum was observed in two skull sides, 1 (0.73%) on the left and 1 (0.78%) on the right side (Fig. 3).

Figure 3 Foramen spinosum (FS) with aberrant anatomical configurations.

Images were captured from the external aspect of the cranial base, from medial to lateral direction. The upper part of the image corresponds to the anterior, the right part to the medial, the left part to the lateral, and the bottom part to the posterior aspect of the cranial base. Image (F) was captured from lateral to medial direction. Image (A) illustrates the absence of the FS. The white arrow points to the foramen spinosum (B, I, J, K, L), a canal-like foramen spinosum (C) and the confluence between foramina (D, E). Specific aberrant anatomical configurations are indicated by black arrows. The “Y-shaped” foramen spinosum had two openings in the middle cranial fossa (indicated by paired white arrows) and a joint opening in the infratemporal fossa (I). FS−foramen spinosum. FO−foramen ovale. FL−foramen lacerum. PSB−pterygospinous bar. SS−spine of the sphenoid bone.

The sphenoid spine was absent unilaterally in all cases, on the right side in two skull halves (1.46%) and on the left side in 1 skull half (0.78%). The position of the FS in relation to the sphenoid spine is depicted in Table 3.

Table 3 Position of foramen spinosum (FS) in relation to the spine of the sphenoid bone.

Position of FS	Right side
(n = 132)	Left side
(n = 125)	Total
(n = 257)	
Anteromedial (%)	44.70	52.80	48.64	
Anterolateral (%)	20.45	20.00	20.23	
Anterior (%)	17.42	12.00	14.79	
Lateral (%)	6.06	5.60	5.84	
Posterolateral (%)	3.79	3.20	3.50	
Medial (%)	1.52	2.40	1.95	
Posteromedial (%)	0.76	0.00	0.39	
Posterior (%)	0.76	0.80	0.78	
Apex of the spine (%)	4.55	3.20	3.89	

No statistically significant differences between the left and right sides were found in any measured parameter.

Discussion

Precisely evaluating anatomical variations in the FS is crucial as it provides an important reference point for various neurovascular surgical procedures in the middle cranial and the infratemporal fossa. Its location renders it useful in endovascular embolization of dural arteriovenous fistulas, meningiomas and recurrent chronic subdural hematomas through the MMA (Mewada et al., 2016; Link et al., 2018; Haldrup et al., 2020). The FS is similarly important in vascular graft surgeries that involve using the MMA as a graft in bypass procedures, such as anastomosis with the petrous part of the internal carotid artery or posterior cerebral artery for treatment of high-flow cervical vascular lesions and tumors in the infratemporal fossa (Ustün et al., 2004, 2006). The principle of early devascularization in meningioma microsurgery includes the identification of feeders and proximal obliteration by coagulation or preoperative embolization (Cvetko & Bosnjak, 2014). In addition, the FS is surgically important during ligation of the middle meningeal vessels that may be torn in the skull base fracture, as well as embolization of the MMA to control bleeding in cases of intractable epistaxis or as a preoperative measure to reduce blood loss during resections of skull base tumours, including upper petroclival lesions (Yu et al., 2016; Shibao, Borghei-Razavi & Yoshida, 2017). Furthermore, the FS is a reference point for the superior petrosal triangle, which is crucial in locating the bony tegmen over the head of the malleus and increases the safety of the surgical procedure (Miller & Pensak, 2003). The FS defines the Glasscock’s posterolateral triangle providing an access point to the cavernous sinus to resect tumors (Melamed et al., 2009) and the lateral triangle of the middle cranial fossa for extracranial-intracranial bypass procedures involving the maxillary artery (Hendricks et al., 2022).

The proximity of the FS to the foramen ovale and trigeminal nerve is of significant clinical relevance. Cannulation of the foramen ovale is employed in percutaneous biopsies targeting cavernous sinus lesions (Messerer et al., 2012), and deep-seated tumors like squamous cell carcinoma, meningioma, and Meckel cave’s lesions (Dresel et al., 1991; Barakos & Dillon, 1992), as well as in intracranial electrode placement for electroencephalographic assesment of temporal lobe seizures (Wieser & Siegel, 1991), and trigeminal rhizotomy procedures performed in individuals suffering from trigeminal neuralgia (Šink et al., 2023). Caution is essential during these procedures, as there is a potential risk of injury to the middle meningeal vessels due to the close proximity and occasional confluence with the foramen ovale (Ginsberg et al., 1994; Berge & Bergman, 2001; Mandavi & Mishra, 2009; Khairnar & Bhusari, 2013; Tewari et al., 2017; Šink et al., 2023). In the present study, the shortest distance between the foramen ovale and the FS was 0.25 mm on the right side, and was 0.72 mm on the left side. The confluence between the foramina was observed in two (0.8%) skull sides. Our measurements are consistent with those reported in the literature (Unver Dogan et al., 2014; Lazarus, Naidoo & Satyapal, 2015; Tewari et al., 2017).

In the present study, the results of the morphometric analysis of the 266 FS were consistent with those reported in other studies conducted on populations of European, American, African and Asian descent (Berlis et al., 1992; Mandavi & Mishra, 2009; Osunwoke et al., 2010; Unver Dogan et al., 2014; Zdilla, Laslo & Cyrus, 2014; Sophia & Kalpana, 2015; Lazarus, Naidoo & Satyapal, 2015; Tewari et al., 2017; Worku & Clarke, 2021; Sugano et al., 2022). However, most existing morphometric studies of FS were limited to measurements of FS length and width (Table 4). The shortest width of the FS, measured in this study, was 0.42 mm on the right side and 0.77 mm on the left side. In addition, we observed a pinhole-like FS in two skull sides. A small or hypoplastic FS may be associated with anomalous origins and trajectories of the MMA (Cvetko & Bosnjak, 2014; Bonasia et al., 2020) and may hinder some surgical procedures involving the middle cranial fossa. On the other hand, in the case of an abnormally enlarged FS, cranial dural arteriovenous fistulas and meningiomas should be considered in the differential diagnosis (Wickbom & Stattin, 1958; Lin et al., 2015).

Table 4 Comparison of foramen spinosum (FS) dimensions between the present and previous studies.

Authors (Year) (Country)	No. of skull sides	Length: mean ± SD (mm)	Width: mean ± SD (mm)	
Right side	Left side	Right side	Left side	
Berlis et al. (1992) (Germany)	120	2.60 ± 0.52	2.07 ± 0.28	
Osunwoke et al. (2010) (Nigeria)	174	2.34 ± 0.05	2.36 ± 0.05	1.66 ± 0.03	1.61 ± 0.03	
Krishnamurthy, Chandra & Rajanna (2013) (India)	100	2.64 ± 0.58	2.44 ± 0.64	2.28 ± 0.54	2.13 ± 0.41	
Unver Dogan et al. (2014) (Turkey)	62	2.90 ± 1.19	2.90 ± 0.67	1.90 ± 0.41	2.01 ± 0.47	
Srimani et al. (2014) (India)	80	2.01 ± 0.31	2.03 ± 0.29	1.65 ± 0.25	1.70 ± 0.19	
Lazarus, Naidoo & Satyapal (2015) (South Africa)	200	2.47 ± 0.73	2.59 ± 0.78	/	/	
Somesh et al. (2015) (India)	164	3.43 ± 0.64	3.34 ± 0.66	2.69 ± 0.49	2.68 ± 0.47	
Sophia & Kalpana (2015) (India)	80	2.37 ± 0.60	2.32 ± 0.67	2.32 ± 0.28	1.73 ± 0.34	
Tewari et al. (2017) (India)	136	2.96 ± 1.13	3.23 ± 0.81	2.17 ± 0.80	2.27 ± 9.17	
Chanda et al. (2019) (India)	60	2.05 ± 1.09	2.05 ± 0.6	1.33 ± 0.9	1.67 ± 0.53	
Katara et al. (2020) (India)	120	3.14 ± 0.53	3.05 ± 0.67	2.68 ± 0.71	2.58 ± 0.65	
Veda et al. (2020) (India)	48	3.04 ± 0.91	2.87 ± 0.95	1.79 ± 0.66	1.66 ± 0.64	
Worku & Clarke (2021) (Ethiopia)	128	3.72 ± 1.33	3.37 ± 1.26	3.3 ± 1.19	2.97 ± 1.9	
Varalakshmi, Nayak & Sangeeta (2022) (India)	180	2.66 ± 0.71	2.58 ± 0.78	1.42 ± 0.32	1.64 ± 0.30	
Present study (2023) (Slovenia)	257	2.45 ± 0.65	2.49 ± 0.61	2.03 ± 0.53	2.08 ± 0.48	
Note:

SD, Standard deviation.

Our observations revealed important variations in the shape of the FS, with the round shape being most frequently identified, followed by oval, irregular, and drop shapes, as shown in Fig. 2 and Table 2. Consistent with findings of previous studies, we found no significant differences in the shape of the FS with regard to skull side (Table 5). The heterogeneity in FS shape may be indicative of pathologies in the middle cranial fossa including benign or malignant primary and secondary lesions in the greater wing of the sphenoid bone. In addition, such variations in the shape of the FS may complicate surgical or other interventional procedures involving transforaminal access to the structures of the middle cranial fossa (Cochinski et al., 2022).

Table 5 Comparison of foramen spinosum (FS) shapes between the present and previous studies.

Authors (Year) (Country)	No. of skull
sides	Round (%)	Oval (%)	Drop shaped (%)	Irregular (%)	Pinhole (%)*	
		Right	Left	Right	Left	Right	Left	Right	Left	Right	Left	
Desai, Saheb & Shepur (2012) (India)	250	52	42	/	6	/	
Rai, Gupta & Rohatgi (2012) (India)	70	57	51.4	34.2	31.4	/	/	2.8	2.8	5.7	8.5	
Krishnamurthy, Chandra & Rajanna (2013) (India)	100	55	40	/	5	/	
Srimani et al. (2014) (India)	80	51.25	30.00	/	7.50	/	
Lazarus, Naidoo & Satyapal (2015) (South Africa)	200	58.5	50.5	36.8	44.9	/	/	/	/	/	/	
Somesh et al. (2015) (India)	164	56.0	51.2	34.1	36.5	/	/	3.6	4.8	6.0	7.3	
Sophia & Kalpana (2015) (India)	80	52.5	30.0	/	12.5	2.5	
Saheb & Havaldar (2017) (India)	600	58	38	/	4	/	
Tewari et al. (2017) (India)	136	55.56	33.33	/	3.96	7.14	
Chanda et al. (2019) (India)	60	53.3	40.0	/	6.7	/	
Katara et al. (2020) (India)	120	56.7	60.0	43.3	40.0	/	/	/	/	/	/	
Veda et al. (2020) (India)	48	71	23	/	2	4	
Worku & Clarke (2021) (Ethiopia)	128	53.12	46.87	31.25	34.37	/	/	6.25	6.25	9.37	12.5	
Sugano et al. (2022) (Brazil)	60	32.8	42.1	12.5	12.5	/	
Varalakshmi, Nayak & Sangeeta (2022) (India)	180	54	47	34	39	/	/	2	4	/	/	
Present study (2023) (Slovenia)	257	63.1	50.0	21.5	35.2	7.7	4.9	7.7	9.8	/	/	
Note:

*The pinhole shape denotes an irregular FS with a very small diameter. We discovered two pinhole-like FS, both on the left side, and classified them as irregular.

The substantial variation in size and shape of the FS across different human populations may be attributed to the effects of genetic diversity on the complex embryological development of the sphenoid bone involving a combination of intramembranous and endochondral ossification processes (Lang, Maier & Schafhauser, 1984; Yanagi, 1987; Nemzek et al., 2000).

The FS originates from the mandibular or first pharyngeal arch within the sphenoid bone. Ossification of the FS commences approximately 8 months postnatally and may require up to 7 years for complete development into its mature bony ring-shaped formation (Yanagi, 1987). Aberrant anatomical configurations of the FS may be attributed to incomplete osteogenesis of the foramen lacerum, petro-sphenoid junction or spheno-squamosal suture (Greig, 1929). Overossification during the developmental process of the sphenoid bone comprising the FS may result in morphologic abnormalities, such as bony spines, tubercles, plates and bars, which may hinder surgical approaches and compress the middle meningeal vessels and nerves, and consequently result in headaches and other clinical complications (Henry et al., 2020; Edvinsson et al., 2020).

The present study observed marginal bony outgrowths of the FS in 8 out of 266 (3.0%) analyzed skull sides. We found 4 FS with marginal bony tubercles (1.5%) and 4 FS with a marginal bony spine (1.5%). Additional 21 (7.9%) FS exhibited irregular marginal morphology due to small outgrowths that did not conform to previously reported classifications (Fig. 3). Marginal irregularities were determined to be non-postmortem, as the edges were smooth.

The FS was absent in 4.38% on the right side, and 3.10% on the left side. Previous studies reported the incidence of unilateral FS to be less than 3% (Rai, Gupta & Rohatgi, 2012; Kulkarni & Nikade, 2013; Sophia & Kalpana, 2015; Somesh et al., 2015). It is believed that the absence of the FS reflects the anomalous formation of the MMA due to the complex embryologic development (Cvetko & Bosnjak, 2014). During development, the third aortic arch serves as the origin of the stapedial artery, which arises from the internal carotid artery and later incorporates into the external carotid artery. The stapedial artery subsequently divides, giving rise to the MMA. Typically, the stapedial artery regresses by the tenth week of development. However, the intricate process of the MMA formation can result in numerous anatomical variations and anastomoses (Tubbs et al., 2015). Possible alternative origins of the MMA include the sphenomaxillary portion of the maxillary artery (Low, 1946), the ophthalmic artery (Gabriele & Bell, 1967; Cvetko & Bosnjak, 2014), the persistent stapedial artery (Altmann, 1947), the pre-petrous, suprasellar or juxtasellar portion of the internal carotid artery (Dilenge & Ascherl, 1980), the basilar artery (Kumar & Mishra, 2012), the ascending pharyngeal artery (Moret et al., 1978) and the posterior inferior cerebellar artery (Kuruvilla et al., 2011). The MMA may also enter the middle cranial fossa through the foramen ovale along with the mandibular nerve (Chandler & Derezinski, 1935). Therefore, variations in morphology and symmetry of the FS are important for the vascular supply of the dura. Consideration should be given to the possible presence of an anomalous MMA in preparation for surgical and endovascular treatment of highly vascularized tumors, such as meningiomas and arterio-venous dural fistulas, in the orbit and the middle cranial fossa (Cvetko & Bosnjak, 2014) as well as dealing with fractures of the base of the skull, epidural hematomas, and surgery of the nerve of the pterygoid canal (Kuruvilla et al., 2011).

In the present study, one FS duplication (0.73%) was observed on the right side (Fig. 3). The unusual position or absence of a typical FS may manipulate the anatomical organization of neurovascular structures passing through the foramen. Duplication of the FS may be associated with the early division of the MMA into anterior and posterior branches (Ginsberg et al., 1994). In addition, the bifurcation may alter blood flow dynamics (Zdilla, Laslo & Cyrus, 2014). We observed one “Y-shaped” FS case, with two openings in the middle cranial fossa and a joint opening in the infratemporal fossa (Fig. 3). The formation of two openings may result from early branching of the MMA (Khan, Asari & Hassan, 2012) or partial confluence with the canalis innominatus, which is normally found between the foramen ovale and the FS to transmit the lesser petrosal nerve (Khairnar & Bhusari, 2013). We also observed confluence between the FS and the foramen lacerum in 2 (0.8%) skull sides, possibly due to incomplete osteogenesis of the foramen lacerum (Greig, 1929). These findings are consistent with those reported in previous studies (Fig. 3 and Table 6) (Berge & Bergman, 2001; Mandavi & Mishra, 2009; Sophia & Kalpana, 2015).

Table 6 Comparison of foramen spinosum (FS) aberrant anatomical configurations between the present and previous studies.

Authors (Year) (Country)	No. of skull sides	Absent FS (%)	Duplicate FS (%)	Confluent FS
with FO (%)	
Right	Left	Right	Left	Right	Left	
Berlis et al. (1992) (Germany)	120	0.8	0	0	
Berge & Bergman (2001) (USA)	184	1.1	0	2.2	
Mandavi & Mishra (2009) (Japan)	624	0.30	2.56	1.60	
Osunwoke et al. (2010) (Nigeria)	174	0	0	0	
Desai, Saheb & Shepur (2012) (India)	250	/	/	/	
Khan, Asari & Hassan (2012) (Malaysia)	50	2.0	2.0	2.0	
Nikolova et al. (2012) (Bulgaria)	959	0.52	/	/	
Rai, Gupta & Rohatgi (2012) (India)	70	2.85	0	2.85	
Kulkarni & Nikade (2013) (India)	200	2.50	0.5	0	
Khairnar & Bhusari (2013) (India)	200	0.5	3.0	1.5	
Krishnamurthy, Chandra & Rajanna (2013) (India)	100	1	0	2	
Srimani et al. (2014) (India)	80	0	3.75	2.5	
Lazarus, Naidoo & Satyapal (2015) (South Africa)	200	2.0	2.5	0	
Somesh et al. (2015) (India)	164	1.22	0	0	
Sophia & Kalpana (2015) (India)	80	2.5	3.75	0	
Saheb & Havaldar (2017) (India)	600	1.1	/	/	
Tewari et al. (2017) (India)	126	0	0	0	
Chanda et al. (2019) (India)	60	3.3	3.3	0	
Javed et al. (2020) (Pakistan)	70	1.43	4.29	/	
Veda et al. (2020) (India)	48	0	16	0	
Kastamoni et al. (2021) (Turkey)	354	0	8.76	0	
Worku & Clarke (2021) (Ethiopia)	128	2.34	0.78	1.56	
Sugano et al. (2022) (Brazil)	60	0	13.3	0	
Present study (2023) (Slovenia)	266	4.38	3.10	0.73	0	0	1.55	
Note:

FS, foramen spinosum; FO, foramen ovale.

The presence of a pterygospinous bar can limit the space between the lateral pterygoid plate and the sphenoid spine and consequently hinder access to the FS and preclude the cannulation of the foramen ovale (Das, Sreepreeti & Gyanaranjan, 2018). It may also be associated with anomalous trajectory and entrapment of the neurovascular structures passing through the FS (Loughner, Larkin & Mahan, 1990; Suazo Galdames, Zavando Matamala & Luiz Smith, 2010). The present study observed 15 complete (5.6%) and 22 incomplete (8.2%) pterygospinous bars in 266 analyzed skull sides. The pterygospinous bar distinctly obstructed the FS in four cases (1.5%) (Fig. 3).

Typically, the spine of the sphenoid bone projects posterior and lateral to the FS. In our study, the FS was anteromedial to the sphenoid spine in 48.6% of cases. Similarly, Mandavi & Mishra (2009) reported the incidence of a normally positioned FS to be roughly 50%. Sophia & Kalpana (2015) observed the normal position only in 30% on the right and 34% on the left side. In the present study, the FS was present at the apex of the sphenoid spine in 3.9% of cases. The spine was broader and flattened at the tip, likely due to the passage of a canal-like FS. Similar findings were reported by a study on 312 dried Japanese human skulls (Mandavi & Mishra, 2009). Changes in the relation between the FS and the sphenoid spine may affect the course of middle meningeal vessels, dural venous sinuses accompanying the MMA, and the nervus spinosus, as well as the chorda tympani and the auriculotemporal nerve (Mandavi & Mishra, 2009; Tewari et al., 2017). Consequently, all aforementioned structures may be damaged during surgical procedures in this area (Cillo, Sinn & Truelson, 2005; Ellwanger & De Campos, 2013).

A major potential complication of surgery in the area of the temporomandibular joint (TMJ) is the possible severance of the MMA (Baur et al., 2014). Lukic, Gluncic & Marusic (2001) reported that abnormal branching of the MMA could potentially lead to significant complications during transantral surgery, radical maxillectomy, lateral approaches to the infratemporal fossa, anterior and medial portions of the skull base, as well as the dorsocaudal approach to the medial skull base.

Additionally, we measured the distance between the FS and the most inferior part of the zygomaticotemporal suture to improve the extracranial localization of the FS. Our results were consistent with those reported by Tewari et al. (2017). We also measured the distance between the FS and the anterior border of the zygomatic root of the temporal bone, as it may serve as a dependable anatomical landmark for guiding the surgeon during navigation through the middle cranial fossa. Peris-Celda et al. (2018) reported that the FS and the foramen ovale would be found within the zygomatic root width in depth in 86% of cases. The distance between the FS and the external aperture of the carotid canal was also measured since the MMA may arise from the cervical segment of the internal carotid artery (Baltsavias, Kumar & Valavanis, 2012). Furthermore, a bypass between the MMA and the petrous section of the internal carotid artery may be created for treating patients with high-flow cervical vascular lesions and infratemporal fossa tumors (Ustün et al., 2004; Kim, Paek & Lee, 2019). External landmarks may help but cannot indicate the precise location of the FS. However, quick and efficient identification of landmarks reduces brain retraction time and associated morbidity (Krayenbühl, Isolan & Al-Mefty, 2008).

We acknowledge a number of limitations in the present study. First, we could not characterize the supporting anthropometric parameters as the sex and age of individuals from whom the skulls were obtained are unknown. Similarly, as there are no clinical or autopsy data, the exact cause of variations observed in the present study is difficult to determine. Accordingly, it is impossible to exclude any potential underlying disease that would cause pathologic changes in the size, shape or anatomical relationships of the skull foramina. Nevertheless, it is generally considered that genetic, nutritional or environmental factors may contribute to such anatomical variations.

Conclusions

Our observations underscore important interindividual differences in the dimensions, shape, and other anatomical variations of the FS, as well as its spatial relationships to surrounding structures of the skull base. These variations in anatomical characteristics should be considered in evaluating pathologies in the middle cranial fossa and planning relevant surgical, radiological and anesthetic interventions in this region.

Supplemental Information

Supplemental Information 1 Raw data.

Morphometric measurements of foramen spinosum. All measurements are in mm. Foramen spinosum position relative to spine. A – anterior. P – posterior. M – medial. L – lateral. Aberrant anatomical configurations of the foramen spinosum.

Click here for additional data file.

We are grateful to Friderik Štendler, Ivan Blažinovič, Stanko Kristl, Sebastijan Krajnc, and Marko Slak for the technical support, and Chiedozie K. Ugwoke for the manuscript proofreading.

Additional Information and Declarations

Competing Interests

Author Contributions

Human Ethics

Data Availability

The authors declare that they have no competing interests.

Žiga Šink conceived and designed the experiments, performed the experiments, analyzed the data, prepared figures and/or tables, authored or reviewed drafts of the article, and approved the final draft.

Nejc Umek conceived and designed the experiments, performed the experiments, analyzed the data, authored or reviewed drafts of the article, methodology, supervision, and approved the final draft.

Erika Cvetko conceived and designed the experiments, performed the experiments, authored or reviewed drafts of the article, methodology, supervision, and approved the final draft.

The following information was supplied relating to ethical approvals (i.e., approving body and any reference numbers):

The National Medical Ethics Committee of the Republic of Slovenia approved the study (0120-459/2018/3).

The following information was supplied regarding data availability:

The measurements and storage information of the specimens are available in the Supplemental File.

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
