# Peer review of "Morphometric and morphologic analysis of the foramen spinosum in the Slovenian population with clinical correlations"

_PeerJ, doi:10.7717/peerj.16559_

## Round 0.1 · original submission · Minor Revisions

We have received four reviews of your paper, and they are very positive about the manuscript. Two recommend accept, one suggests minor revisions and a fourth asks for major revisions. However, in sum I deem this paper requiring only minor revisions, and these are limited to the figures. 1) Please either indicate on one of the images or provide a new figure that illustrates the measurements collected for the study. 2) Provide arrows or other indicators on the figures to help the reader understand the anatomy that is described in the legends and main body text.

The manuscript is very clear and well-written, but double-check all portions of the manuscript including the references as typographical errors are easier to correct now than at the page proof stage.

·

Basic reporting

A well-written and comprehensive study on the anatomy and morphometric features of the foramen spinosum in the Slovenian population.

Experimental design

A well-designed study providing particular details on specimen evaluation.

Validity of the findings

The findings are useful and can help us better understand the anatomy under study.

·

Basic reporting

This study by Sink et al explored 266 middle fossas (126 complete skulls, and 15 skull halves) to identify anatomic size and proximity of foramen spinosum to neighboring bony landmarks in a cohort of donated dry Slovenian skulls. This is well written and organized. The authors crafted a thoughtful introduction and discussion section, and overall the manuscript is well assembled. The imaging is of good quality, and along with the tables, the figures/tables are easy to follow. The manuscript is thoroughly referenced, and the methods seem complete.

I would suggest adding arrows or some form of intra-image tool to highlight the descriptive points being made in each image for figures 1 and 2, particularly to aid those readers in training or for those with minimal neuroanatomic experience.

Experimental design

The experimental design and execution of the manuscript is good. The authors noted that they had received guidance to split anatomic exploration of foramen spinosum and ovale into two distinct manuscript. From my perspective, it may have been nice to including them into the same manuscript due to close anatomic proximity and therefore high surgical relevance. However, it is not unreasonable to split these findings.

As the authors mention, there have been numerous other studies describing similar spinosum structure characteristics, but I agree that similar studies across distinct patient populations is often of value. Of course, having true cadaveric/imaging validation of their findings would also add value, but is likely outside the scope of this study.

Validity of the findings

Similar to above, other studies have performed similar investigations, however this study does study a different patient cohort and is particularly large scope. In addition the manuscript is well written and I think nicely contributed to the over all literature, as well reviewing the previous anatomic cohort studies - giving some credence to this being a relatively encompassing and definitive article.

·

Basic reporting

Abstract is well structured, Better to add Key words sub heading also in Abstract. Introduction is appropriate, elaborated nicely about subject of Foramen spinosum. Material and methods are fine. Results are explained well in tables. Discussion is written very elaborative. Conclusion is fine.

Article is acceptable after mentioning key words on Abstract.

Experimental design

Study design is appropriate.

Validity of the findings

It is good. Findings are validated.

Additional comments

Acceptable.

·

Basic reporting

The study planning is good and the manuscript writing and its significance is good.
However the incidence of patients undergoing transpinosal approach and its complication incidence in Slovenian population is not mentioned.

Experimental design

1. It is better if the images of how the measurement using Vernier caliper added.
2. The authors have used half skulls and justification of both side significance have not been mentioned.

Validity of the findings

The study have been done on unidentified sex, though there are significant male female dimensional findings it is better if the authors identify the sex of the skull and differentiate male and female findings.

Additional comments

The study only concentrated the surgical approach through FS, but other clinical correlations related to different shapes and absent FS have not been discussed

---

## Round 0.2 · accepted · Accept

Thank you very much for your revised manuscript and edits to the figures. Your comments satisfactorily responds to the reviewers' concerns. I have marked the revision for acceptance.